# Endothelial Dysfunction: An Intermediate Clinical Feature between Urolithiasis and Cardiovascular Diseases

**DOI:** 10.3390/ijms23020912

**Published:** 2022-01-14

**Authors:** Javier Saenz-Medina, Mercedes Muñoz, Claudia Rodriguez, Ana Sanchez, Cristina Contreras, Joaquín Carballido-Rodríguez, Dolores Prieto

**Affiliations:** 1Department of Urology, Puerta de Hierro-Majadahonda University Hospital, 28222 Majadahonda, Spain; 2Department of Medical Specialities and Public Health, Faculty of Health Sciences, King Juan Carlos University, 28933 Móstoles, Spain; 3Department of Physiology, Pharmacy Faculty, Complutense University, 28040 Madrid, Spain; mmpicos@ucm.es (M.M.); claudrod@ucm.es (C.R.); aasanche@ucm.es (A.S.); criscont@ucm.es (C.C.); dprieto@ucm.es (D.P.); 4Department of Urology, Puerta de Hierro-Majadahonda University Hospital, Autonoma University, 08193 Bellaterra, Spain; carballidojoaquin@gmail.com

**Keywords:** endothelial dysfunction, oxidative stress, urolithiasis, kidney stones

## Abstract

An epidemiological relationship between urolithiasis and cardiovascular diseases has extensively been reported. Endothelial dysfunction is an early pathogenic event in cardiovascular diseases and has been associated with oxidative stress and low chronic inflammation in hypertension, coronary heart disease, stroke or the vascular complications of diabetes and obesity. The aim of this study is to summarize the current knowledge about the pathogenic mechanisms of urolithiasis in relation to the development of endothelial dysfunction and cardiovascular morbidities. Methods: A non-systematic review has been performed mixing the terms “urolithiasis”, “kidney stone” or “nephrolithiasis” with “cardiovascular disease”, “myocardial infarction”, “stroke”, or “endothelial dysfunction”. Results: Patients with nephrolithiasis develop a higher incidence of cardiovascular disease with a relative risk estimated between 1.20 and 1.24 and also develop a higher vascular disease risk scores. Analyses of subgroups have rendered inconclusive results regarding gender or age. Endothelial dysfunction has also been strongly associated with urolithiasis in clinical studies, although no systemic serum markers of endothelial dysfunction, inflammation or oxidative stress could be clearly related. Analysis of urine composition of lithiasic patients also detected a higher expression of proteins related to cardiovascular disease. Experimental models of hyperoxaluria have also found elevation of serum endothelial dysfunction markers. Conclusions: Endothelial dysfunction has been strongly associated with urolithiasis and based on the experimental evidence, should be considered as an intermediate and changeable feature between urolithiasis and cardiovascular diseases. Oxidative stress, a key pathogenic factor in the development of endothelial dysfunction has been also pointed out as an important factor of lithogenesis. Special attention must be paid to cardiovascular morbidities associated with urolithiasis in order to take advantage of pleiotropic effects of statins, angiotensin receptor blockers and allopurinol.

## 1. Introduction

Nephrolithiasis is a worldwide public health problem affecting nowadays 5% to 9% of the European population and almost 12% of the North American people. It has high recurrence rates reaching 60% of the cases in 10 years. Consequences of nephrolithiasis range from the appearance of obstructive uropathy, frequently resulting in loss of work, need of hospitalization, or surgery, to kidney disease in the most severe cases. Budget-impact analyses based on 65 million population (France) have established the annual cost of urolithiasis in €590 million [1].

It has recently been suggested a significantly increased risk of kidney stone development secondary to diet factors and has been associated with pathologies such as obesity, metabolic syndrome or diabetes [2]. These alterations are also associated with chronic kidney disease (CKD) and epidemiological data indicate that obesity is an independent risk factor for the development of chronic kidney injury [3]. 

Endothelial dysfunction (ED) is an early pathogenic feature of cardiovascular morbidities, consisting of impaired vasodilation, angiogenesis and barrier function [4]. It has been related to metabolic diseases such as diabetes mellitus, obesity or metabolic syndrome; all of them also associated with cardiovascular diseases [5]. The key role of oxidative stress and inflammation in the pathogenesis of ED is well established [4].

Recently, ED has been linked to urolithiasis through clinical and experimental studies [6,7,8,9]. The aim of this review is to summarize the current knowledge and the latest findings on the pathogenic mechanisms underlying endothelial dysfunction in urolithiasis, paying special attention to the role of the reactive oxygen species (ROS) as an underlying mechanism.

## 2. Literature Search Methodology

A non-systematic review of the experimental evidence included in electronic databases, such as PubMed, Science Direct, and Scopus, was performed from 2000 to today. The keywords were “kidney stone” or “urolithiasis”, or “nephrolithiasis”, or “renal calculi”, or “renal stone”, or “endothelial dysfunction”, or “oxidative stress”, or “reactive oxygen species”, or “cardiovascular diseases”, or “stroke”, or “myocardial infarction”, or “coronary heart disease”. The resulting articles were combined according to the title of the following sections. In a more specific way, the clinical studies in which association between urolithiasis and cardiovascular diseases have been studied have been summarized (Table 1). 

## 3. ROS and Renal Oxidative Stress Sources

Reactive oxygen species (ROS) are a group of molecules with different effects on cellular function that include molecular oxygen derivatives such as superoxide anion (O_2_^−^), hydroxyl radical (OH), hydrogen peroxide (H_2_O_2_), peroxynitrite (OONO^−^), and hypochlorous acid (HOCl). When ROS levels exceed the cellular antioxidant capacity, they cause cell damage oxidizing DNA, proteins, carbohydrates or lipids. This phenomenon is called oxidative stress (OS), a common pathogenic mechanism underlying cardiovascular diseases, neuroregenerative disorders, inflammation and cancer. ROS have also been recognized as signaling molecules that mediate the induction of host defense genes, activation of transcription factors, phosphorylation of kinases and mobilization of ion transport systems [10,11].

ROS can be formed by different sources including mitochondrial respiratory enzymes, nicotinamide adenine dinucleotide phosphate (NADPH) oxidase, the uncoupling of endothelial nitric oxide synthase (eNOS), or the enzymes xanthine oxidase, lipoxygenase and cyclooxygenase (COX) [11,12].

Mitochondria are a major source of ROS generation and render 80% of basal O_2_^−^ by reducing O_2_ consumed by cells. ROS are natural byproducts of mitochondrial respiration; mitochondrial O_2_^−^ is usually scavenged by antioxidant systems such as mitochondrial manganese superoxide dismutase (MnSOD) and glutathione peroxidase. Damaged or dysfunctional mitochondria overproduce O_2_^−^-triggering damaging reactions through production of H_2_O_2_, altering ATP synthesis, dysregulating Ca^2+^ or inducing permeability transition [10]. 

NADPH oxidases, along with mitochondria, are the major producers of superoxide and are the principal sources of abnormal signaling. They catalyze the reduction of molecular O_2_ using NADPH as an electron donor to produce O_2_^−^. Nox family of NADPH oxidases is composed of Nox1, Nox2 (previously identified as gp91phox), Nox3, Nox4, Nox5, Duox1 and Duox2. All Noxes are trans membrane proteins that transport electrons across to reduce O_2_ to O_2_^−^. Nox1, Nox2, Nox4 and Nox5 have been identified in vascular tissue, while Nox4 is highly expressed in both vascular systems and kidneys, being, along with mitochondria, a major source of ROS generation in the kidney [13,14,15]. Nox-derived ROS are implicated in the signaling of physiological processes such as cell proliferation, interaction with the immune system and regulation of the vascular tone [14,16,17]. In the kidney, NADPH oxidases regulate different renal functions including glucose transport, gluconeogenesis, tubuloglomerular feedback, renal hemodynamics and electrolyte transport [14]. NADPH oxidase expression has been identified in vascular endothelial and smooth muscle cells, macrophages and platelets, particularly in pathological states [11,17,18]. O_2_^−^ production derived from Nox is enhanced by several pathophysiological conditions such as tumor necrosis factor (TNFα) expression, integrin ligation, diabetes, or oxidized LDL [12].

Dysregulated or uncoupled NO synthase (NOS) is another source of ROS production in metabolic and cardiovascular diseases [19]. Endothelial NOS (eNOS) is uncoupled by the reduced availability of cofactors such as BH4 that are oxidized by peroxynitrite (ONOO^−^), formed by the reaction of NO with O_2_^−^ under conditions of enhanced O_2_^−^ production [20]. Smoking, hypertension, diabetes, ischemia-reperfusion injury, and coronary artery disease have been associated with a reduced bioavailability of NOS cofactors and eNOS uncoupling.

Xanthine oxidase (XO) catalyzes the conversion of hypoxanthine to xanthine and xanthine to uric acid, rendering O_2_^−^ as a byproduct of the reaction, and also produces H_2_O_2_ under low O_2_ conditions [21]. Oxidative stress enhances the conversion of xanthine dehydrogenase to XO, thus increasing ROS levels and oxidative stress in a feed-forward mechanism. XO is a major contributor to oxidative stress in ischemia-reperfusion [22].

Since ROS overproduction leads to oxidative stress, cells have antioxidants systems that rapidly break down ROS to less reactive or non-reactive products, including glutathione and thioredoxin redox circuits, and antioxidant enzymes such as superoxide dismutase (SOD), catalase and glutathione peroxidase (Gpx). SODs convert O_2_^−^ to H_2_O_2_ and there are three isoforms: (1) SOD1, a CuZn-SOD found in the cytoplasm that suppresses scavenging of NO by O_2_^−^ and regulates angiogenesis and vasomotor tone; (2) SOD2, a mitochondrial Mn-SOD located in the mitochondrial matrix and readily inactivated by peroxynitrite, whose deletion induces perinatal lethality due to cardiomyopathy; and (3) SOD3, secreted and then tethered to the outer plasma membrane, is particularly important in the cardiovascular system due to its high expression in blood vessels, lung and heart [12,17].

Catalase is a heme-containing enzyme located in cytosol and peroxisomes that converts H_2_O_2_ to water and molecular oxygen. GPx and peroxiredoxins (Prx) reduce H_2_O_2_ to water by oxidizing GSH and thioredoxin (Trx) and are found in many cellular regions wherein and they both reduce peroxide and regulate signaling mechanisms [12,17].

In response to oxidative stress, cells activate redox-sensitive transcription factors such as the nuclear factor (erythroid 2–related) factor 2 (Nrf2), and thus transcription of protective antioxidant genes including catalase, SOD, Gpx, Prx, and Trx [23].

Enhanced ROS generation and oxidative stress have extensively been associated with cardiovascular and kidney diseases. Research data point out that OS must be considered either as a primary or a secondary cause for many cardiovascular diseases (CVDs), acting as a trigger of atherosclerosis, in which endothelial inflammation is followed by the recruitment of inflammatory cells. Thus, oxidative stress has been implicated in the pathogenesis of atherosclerosis, hypertension, cardiomyopathy, ischemia, ictus, congestive heart failure and in the vascular complications of diabetes and obesity [4,11,24]. Likewise, oxidative stress affects some kidney structures such as glomeruli, tubules, and renal blood vessels inducing the recruitment of inflammatory cells and the proinflammatory cytokine (TNFα), and transcription factors (NF-κB), which results in an inflammatory stage and later fibrosis that impair kidney function [12,14,15,25,26]. 

## 4. Oxidative Stress and Endothelial Dysfunction 

The vascular endothelium acts as an interphase that regulates the passage of nutrients, hormones and macromolecules from the blood to the surrounding tissue, but also ensures the fluidity of blood and contributes to homeostasis. Thus, the endothelium is now considered as the largest endocrine, autocrine and paracrine organ of the body, that secrets vasoactive and trophic mediators which regulate vascular tone, endothelial and vascular smooth muscle cell proliferation, coagulation, inflammation and permeability [27,28]. In response to flow-induced shear stress or to chemical signals, endothelial cells release vasodilator, antiaggregant and anti-inflammatory factors such as nitric oxide (NO), cyclooxygenase (COX)-derived prostacyclin (PgI_2_) or endothelium-derived hyperpolarizing factors (EDHFs), and on the other hand, vasoconstrictor, proaggregant and proliferative mediators such as thromboxane A_2_ (TXA_2_), ROS and endothelin 1 (ET-1) [29]. Endothelial cells also modulate the immune reaction by the recruitment of inflammatory cells through the induction of leucocyte adhesion molecules, cytokines or ROS [4,30].

Endothelial dysfunction (ED) consists of an impaired vasodilator response and altered angiogenesis and barrier function, along with an elevated expression of pro-inflammatory and pro-thrombotic factors, because of an endothelial maladaptation to mechanical, metabolic or oxidative stresses. ED is considered to be the first stage of vascular disorders and atherogenesis and it is considered to occur in the preclinical phase of vascular disease, predisposing it to complications [27,31].

The key role of oxidative stress and inflammation in the pathogenesis of ED is well established. Despite several factors that can compromise the availability of NO, which protects the vascular wall from the events leading to atherosclerosis, one of the primary causes of ED is oxidative stress in many metabolic and cardiovascular diseases. NO produced by eNOS interacts and is rapidly inactivated, by O_2_^−^-producing peroxynitrite, a highly potent oxidant and toxic radical that injures DNA, proteins and lipids, and also inactivates PgI_2_, thus contributing to impaired PgI_2_-mediated vasodilation and antiaggregation. NADPH oxidase is considered to be a major source of vascular ROS generation under some pathological conditions. Augmented ROS production activates the oxidative stress-sensitive nuclear transcription factor (NF-κB) that directly up-regulates NADPH oxidase and regulates the expression of genes encoding adhesion molecules, COX-2, and pro-inflammatory cytokines TNFα, IL-6 and C-reactive protein (CRP), which in turn may activate NADPH oxidase and ROS-generation, impairing endothelial function [4,27].

Oxidative stress is considered to be the main trigger factor for the vascular complications in diabetes Mellitus, including diabetic nephropathy which leads to CKD. Increased ROS production is mainly derived from mitochondria and NADPH oxidase, and also from the polyol pathway, uncoupling NOS, xanthine oxidase or advanced glycation, which in turn results in a thickening of the glomeruli basement membrane, mesangial expansion, hypertrophy and podocyte loss [12,32]. The relationship between diabetes mellitus and ED is also well established. Microangiopathy is one of the main complications of diabetes and is a key factor for the development of retinopathy, nephropathy and diabetic foot. In type 1 diabetes, ED is a consequence of hyperglycemia-associated oxidative stress, while in type 2 diabetes, other factors, such as dyslipidemia, hyperinsulinemia and an abnormal release of adipose tissue-derived adipokines, are also associated [12,14,28].

Overweight and obesity-associated ED was first reported in clinical studies demonstrating blunted increases in leg blood flow in response to the administration of endothelium-dependent agonists in obese patients with an augmented body mass index (BMI) [33]. The vasculature and endothelial cells are affected by nutrient overload, suggesting that the susceptibility to oxidative stress, inflammation and insulin resistance is greater than in other tissue types [5]. Altered production of adipokines and the elevated levels of free fatty acids (FFA) induce OS, leading to a reduced NO availability, an imbalance between vasodilator and vasoconstrictor prostanoids, impairment of EDHF–mediated responses, and elevation of vasoconstrictor and proatherogenic factors such as ET-1. Oxidative stress and inflammation play an essential role in the development of ED. Hypertrophy of adipose tissue leads to a proinflammatory phenotype marked by an increased ROS production and an elevation of circulating of proinflammatory cytokines as IL-6, TNFα, monocyte chemoattractant protein-1 (MCP-1) and plasminogen activator inhibitor type 1 (PAI-1) [4]. Vascular oxidative stress derived from COX-2, Nox1 and Nox2 underlies kidney ED and inflammation in obesity [15,26,34]. 

Other traditional cardiovascular disease-associated factors, such as hyperlipidemia, smoking or hypertension, have also been associated with ED and oxidative stress. Furthermore, new clinical and environmental features have been also pointed out. Mental stress has been shown to activate the immune system and lead to adverse cardiovascular effects [35]. Chronic autoimmune diseases such as lupus erythematosus, rheumatoid arthritis or severe psoriasis also increase cardiovascular risk through ED [36,37,38,39]. Aging is also associated with arterial stiffness and ED, as a result, the incidence and frequency of cardiovascular complications is higher among the elderly [40]. COVID 2019 has also been pointed out as an endothelial disease, and the consequent endotheliopathy is responsible for inflammation, cytokine storm, coagulopathy and oxidative stress [41].

## 5. Nephrolithiasis, Oxidative Stress and Inflammation 

### 5.1. Pathogenesis of Urolithiasis

Nephrolithiasis refers to the presence of stones originated in the kidney; they are formed as a result of crystallization of substances excreted in urine. It affects approximately 10% of the population and has a high rate of recurrence. Calcium stones constitute approximately 80% of the total, while uric acid represents 15%. Other less frequent compositions are cystine, or struvite. 

Four different possible mechanisms of stone formation have been identified: (1) growth over Randall’s plaque, mostly seen in idiopathic calcium oxalate (CaOx) stone formers, in which most of the calculi are formed on Randall plaque; (2) growth over Bellini duct plugs, mostly seen in CaOx and calcium phosphate (CaP) formers; (3) formation of microlithiasis within inner medullary collecting ducts, only observed in cystinuria; and (4) formation in free solution within renal calyces or renal collecting system, observed in CaOx stone in patients with primary hyperoxaluria or obesity-bypass, brushite and hydroxyapatite stone formers. Oxidative stress has been implicated in the two first mechanisms through an endothelial injury as an important step for the stone formation [42].

In most idiopathic stone formers, an initial deposition of CaP on the basement membrane of the loops of Henle appears to form a nidus for subsequent calcium stone formation. Underlying renal cell injury increases the likelihood of crystal attachment to renal epithelial cells. Fragments of injured cells could promote a stone nucleus over the aggregation of crystals will form the kidney stone [43].

Tissue cultures in which renal epithelial or other cells are exposed to Ox and/or CaOx and or CaP crystals, have provided evidence that crystals bind rapidly to the surface of the epithelial cells, followed by crystal endocytosis into the cells. An increased expression of specific encoding genes, transcriptional activators, matrix regulators, growth factors and inflammatory and anti-inflammatory molecules such as MCP-1, osteopontin (OPN), prostaglandin E2, osteonectin and fibronectin is triggered. These molecules induce inflammation and fibrosis, as well as the modulation of biomineralization [43].

Renal cells exposed to CaOx crystals have been reported to secrete O_2_^−^ and antioxidants and free radical scavengers could ameliorate cellular injury. Thus, NADPH oxidase inhibitors and ROS catalytic enzymes such as catalase or SOD decrease the expression of OPN or MCP-1 [44,45]. Mitochondria, as the most common source of ROS generation, have been identified as a major site of CaOx crystal-induced O_2_^−^ and glutation depletion in renal epithelial cells. Furthermore, isolated mitochondria responded to CaOx exposure by the accumulation of ROS, lipid peroxides and oxidized thiol proteins. Citrate addition to the culture medium was associated with an increase in glutathione, and decreased production of ROS and 8-isoprostane, improving cell viability [46].

In animal models of hyperoxaluria and CaOx deposition, the oxidative stress-mediated inflammatory response by the renal cells to the deposition of CaOx crystals has also been proven [47]. Furthermore, in a recent study performed by our group, oxidative stress associated with Nox1 upregulation along with Nox4 down regulation and an inflammatory response mediated by an elevation of TNFα, COX-2, NF-κB, MCP-1, and OPN have been demonstrated in the kidney of hyperoxaluric rats [48]. Other studies have demonstrated that inhibiting the renin-angiotensin system [49] or decreasing Nox1 and p22^phox^ subunit expression through the administration of atorvastatin inhibits NADPH oxidase-derived OS, inflammation and crystal deposition in rats with experimentally induced hyperoxaluria [50].

### 5.2. Chronic Low Inflammation Diseases and Urolithiasis 

Epidemiological studies have provided evidence for the association between metabolic and cardiovascular disorders and nephrolithiasis. Furthermore, diet factors have been pointed out as the cause of a rising incidence of urolithiasis in industrialized countries from approximately 3% in the 70s to a current prevalence of about 9% [51,52].

The risk of urolithiasis in patients with metabolic syndrome when compared with healthy people has been estimated with an odds ratio between 1.5 and 2.2 [2,53]. In the course of obesity, wherein an inflammatory status is present, oxidative stress is secondary to lipotoxicity. In most tissues lipotoxicity consists of a mitochondrial dysfunction triggered by the production of toxic metabolites from the oxidation of free fatty acids (FFA), leading to cell apoptosis. The kidney, as a target organ of lipotoxicity, exhibits both glomerular and renal tubule lesions (mostly proximal tubule). Kidney damage mechanisms in obesity are both lipotoxic and non-lipotoxic. Renal injury by lipotoxicity occurs due to intracellular FFA excess, whose abnormal mitochondrial metabolism increases the production of metabolites such as ceramide, or diacylglycerol. The latter ROS production and oxidative stress, that damages cell organelles, alters intracellular signal mechanisms, releases pro-inflammatory factors and lipid-induced apoptosis [54].

Epidemiological studies have also been associated with diabetes and urolithiasis. A cross sectional study over more than 200,000 US participants determined a higher risk of urolithiasis in diabetic patients with an RR between 1.31 (95% CI 1.11–1.54) in men, and 1.67 (95% CI 1.28–2.20) in younger women [55]. Conversely, kidney stone formers had a higher prevalence of diabetes mellitus (OR 1.475; 95% CI 1.283–1.696) than non-stone formers in a cross sectional study based on a self–reported questionnaire over 23,349 patients [56]. A case-control study of the electronic data of 3561 nephrolithiasis cases also showed a higher prevalence of diabetes in nephrolithiasis patients. Concerning stone composition, more uric acid stone formers were diabetic than other stone former types. This feature is explained because diabetes type II patients have a low urine pH [57].

Hyperglycemia and glycosuria have been related to alterations in renal handling of calcium, phosphorus and uric acid. Increase in calcium and phosphorus excretion and augmented oxalate and uric acid excretion when poor glycemic control is developed, have been reported in diabetic patients. All of these are risk factors for the development of nephrolithiasis in diabetic patients [58,59,60].

Furthermore, hyperglycemia activates a particular metabolic route that involves diacylglycerol, protein kinase C and NADPH oxidase, culminating in ROS. The increased ROS generation leads to oxidative stress, inducing several cellular changes [61]. High glucose-induced injury and inflammation leads apoptosis and necroptosis that may result in tissue injuries in the heart, retina, nervous system and kidneys. It is believed that diabetes complications are the consequence of persistent hyperglycemia-induced low-grade inflammation and cellular dead [62].

Concerning cardiovascular morbidity, lithiasic patients also develop hypertension more frequently than non-stone formers. Multivariate-adjusted OR of hypertension in lithiasic patients has been estimated between 1.24 and 1.96. Furthermore, female lithiasic patients show higher risk of hypertension and coronary heart disease (CHD) than men (1.43 vs. 1.31, *p* = 0.01) [63]. In this respect, hypertensive patients also develop a higher risk of nephrolithiasis (OR = 2.11 95% CI 1.17–3.81), and the study of Kim et al. showed hypertension as an independent factor of stone recurrence [64]. Animal models of hypertensive rats were found to be more susceptible to developing nephrolithiasis than normotensive ones [65].

Hypertension has also been related to oxidative stress. ROS production in vessels, as well as other organs, including heart, kidneys and brain participate in blood pressure regulation. Additionally, all vascular cell types produce ROS, including endothelial, smooth muscle, adventitial fibroblasts and perivascular adipocytes. The causal relationship between ROS and hypertension probably occurs at vascular level, where oxidative stress promotes ED [4,27,66]. Redox signaling in the central nervous system is also important in neuronal control of blood pressure, and renal oxidative stress is associated with glomerular damage, proteinuria, alteration in ions transport, volume retention and nephron loss, all of them influent in the development of hypertension [67]. Plasma levels of oxidative markers are increased in patients with hypertension. ROS production is increased in vascular smooth muscle cells from resistance arteries of hypertensive patients, associated with the up-regulation of NADPH oxidase. Decreased antioxidant capacity also contributes to OS in patients with hypertension. Hypertensive patients have reduced activity and decreased content of antioxidants enzymes, including SOD, glutathione peroxidase and catalase [68,69,70].

## 6. Endothelial Dysfunction: Clinical Implications, Detection and Correction

Current experimental evidence suggests that endothelial function is a reliable, early predictor of cardiovascular mortality, encompassing production of the different endothelium-derived messengers that help to control vascular tone, blood flow, immune cell and platelet activity/adhesion, which also correlates with classical markers of inflammation, obesity and cardiovascular risk such as CRP, adiponectin and brain natriuretic peptide (BNP) [71,72,73]. Endothelial dysfunction is considered as a marker of the effect of damage on the arterial wall and its intrinsic capacity of repair. Classic cardiovascular risk factors such as hypertension [74], hypercholesterolemia [75], diabetes mellitus [76] and chronic smoking [77] are all associated with ED, and the presence of more than one produces synergistic effects on endothelial function [78]. 

ED is the key factor for the initiation and progression of atherosclerosis [74,75,79]. An impaired endothelium-dependent vasodilation has been evidenced in the presence of atherosclerosis, and different studies have confirmed that ED is present in the preclinical stage of atherosclerosis. Since most cardiovascular diseases are either related or a direct consequence of atherosclerosis, ED is an early predictor of subsequent cardiovascular events or mortality [80,81]. Impaired endothelial function has also been demonstrated in patients with peripheral arterial occlusive [82,83], coronary artery disease [84] or heart failure [85]. ED is also clearly associated with oxidative stress [86], which is another feature in the development of atherosclerosis [87,88,89].

A correlation between ED and coronary risk factors in healthy subjects has also been shown, suggesting the importance of coronary and peripheral measurement of ED in these individuals [90,91]. Conversely, large-scale trials have suggested that ED has no significant prognostic value in cohort studies of mostly healthy subjects [92], and is not an independent predictor of cardiovascular events in individuals with an intermediate cardiovascular risk [93]. However, in patients at high risk of cardiovascular disease, peripheral ED is significantly correlated with impending cardiovascular events [94].

The brain is also an important target for ED. A metanalysis based on 4816 patients affected by ischemic strokes, reported higher levels of ED markers than non-stroke individuals (E-Selectin, P-Selectin, intercellular cell adhesion molecule, and vascular cell adhesion molecule 1 VCAM-1) [95]. Furthermore, higher OS serum markers in patients with myocardial infarction (MI) and stroke have been reported [96], suggesting an important contribution of an imbalanced redox system to the etiology of mainly fatal MI and stroke events.

ED can be assessed by inducing endothelial vasodilation via invasive and non-invasive methods. Infusion of acetylcholine induces endothelium-dependent dilation in healthy coronary arteries and vasoconstriction in the presence of endothelial damage. The coronary artery diameter can be compared by quantitative angiography or by intracoronary Doppler ultrasound [97]. Venous occlusion pletismography has also been used to measure vasomotor responses of forearm resistance vessels during infusion of acetylcholine into the brachial artery [82].

A non-invasive method can be also used to assess endothelial function. High-resolution ultrasound is used to measure the brachial artery in response to reactive hyperemia. Reactive hyperemia induces increased blood flow and shear stress, stimulating NO release and flow-mediated dilation that can be quantified as an index of vasomotor function [98]. Ex vivo assessment of ED is also possible through different techniques. The isometric tension method, described in human vessels, obtained in by-pass surgery. [99], or the identification of arteriosclerotic lesions by Raman spectroscopy [100] are examples of this.

Later stages of atherosclerosis result in arterial stiffness that can be quantified by measuring the pulse wave velocity through applanation tonometry, Doppler ultrasound [101,102], or magnetic resonance imaging [103,104,105]. Measurement of intima-media thickness [92] has been also considered, since it positively correlates with the levels of oxidized thiols and early atherosclerosis [106].

In addition, systemic peripheral blood markers have been used to detect ED, and they have been related to peripheral artery disease. Dymethil arginines are endogenous analogues of L-arginine. They include asymmetric dimethylarginine (ADMA), which acts as an inhibitor of NO synthesis and is associated with ED, and SDMA that may compromise NO formation by limiting the cellular availability of L-arginine. ADMA acts as a predictor of total mortality in patients with peripheral artery disease, and is related to disease severity [107].

CRP, a biomarker of significant inflammation, has been also used as a systemic marker of ED. A high-sensitivity CRP assay that can detect low-grade chronic inflammation has been developed and can be applied to predict coronary heart disease (CHD) in asymptomatic patients [108].

New molecular imaging methods have been developed, in recent years, for the detection of areas of dysfunctional endothelium. The infusion of gaseous microbubbles linked to specific antibodies (e.g against P-selectin, VCAM-1), detected by ultrasound [109], in situ generated microbubbles from chemical reactions [110], L-band electron paramagnetic resonance spectroscopy for the detection of areas of dysfunctional endothelium [111], or the determination of the glycocalix by intravital microscopy by orthogonal polarization [112], are examples of the latest advances.

Different strategies have been proved to be effective in restoring ED. These include lipid-lowering therapy, angiotensin-converting enzyme inhibitors, antioxidants, or therapies based on reducing hyperglycemia, diet and exercise [113]. 

Treatment with statins has been shown to improve endothelial-mediated responses in coronary arteries of patients with atherosclerosis [114]. The clinical benefits observed with statins appear to exceed the benefits expected from the reduction of cholesterol, suggesting that statins could develop non-lipid effects, in which the improvement of ED may be involved. These include an increased availability of eNOS, and inhibition of the activity of Rho/Rho signaling, a reduction of circulating levels of the adhesion molecules P selectin and intercellular cell adhesion molecule-1 (ICAM-1) [115]

Large-scale clinical trials, performed to investigate the impact of antioxidant therapy on clinical end-points and the progression of atherosclerosis, have not found beneficial effects on cardiovascular outcome [116]. The reasons for the disappointing results have pointed out the need of measuring the antioxidant status of each subject in order to establish if there is a pre-existing antioxidant deficiency [117].

Clinical studies have demonstrated that Angiotensin receptor blockers show pleiotropic/off target effects beyond the lowering of arterial pressure. Li et al. [118] evidenced, through a metanalysis, that Angiotensin receptor blockers improved peripheral ED, measured by flow-mediated dilation. Angiotensin-converting enzyme is located on the luminal surface of the endothelial cells. Conversion of Ang I in to Ang II and binding to the AT_1_ receptors induces vasoconstriction, pro-oxidizing and antifibrinolytic effects [119]. Ang II also stimulates the ET converting enzyme degrading big ET thus producing ET-1 an important and potent vasoconstriction in blood vessels [120,121]. Angiotensin receptor blockers have been associated with protective effects on endothelial cells, inducing significant reduction of oxidative stress, apoptotic markers, and reduction of cytokine and inflammatory expression [122].

## 7. Endothelial Dysfunction: A Clinical Feature Highly Associated to Nephrolithiasis

Urolithiasis and cardiovascular disorders have extensively been related through epidemiological studies (Table 1). Three metanalysis have been reported at the time [123,124,125]. All of them have observed a significant association between CAD and urolithiasis with RR between 1.20 and 1.24. Two of them (Peng and Liu) have also reported a significant association of lithiasis with stroke, and more specifically with MI and with coronary revascularization. Analysis of subgroups observed inconclusive results at this point; Peng et al. observed a higher risk of CHD in men (RR: 1.23; 95% CI: 1.02–1.49) than in women (RR: 1.01; 95% CI: 0.92–1.11), although women showed higher risk for stroke (RR: 1.12; 95% CI: 1.03–1.21) than men (RR: 1.03 95%; CI: 0.98–1.09), and for MI (RR: 1.37 95% CI: 1.13–1.67 vs. 1.01 95% CI: 0.92–1.11–1.49). Likewise, a significantly higher risk of CHD has been reported in females (RR: 1.43; 95% CI: 1.12–1.82) compared with men, 95% (RR: 1.14; CI: 0.98–1.38) [125], and in the same way, Liu [124] observed an increased risk for MI in women (HR: 1.49; CI: 1.21–1.82) compared with men, in which no association was observed (HR: 1.15; CI: 0.89–1.50). In addition to these studies, subpopulations of American and Asian lithiasic patients were more susceptible to CHD and stroke [123].

Kim et al. have reported the most recent study, not included in any of three metanalysis, for the evaluation of the association between nephrolithiasis and stroke. Results have shown a higher risk of ischemic stroke in nephrolithiasis patients (RR: 1.13; 95% CI 1.06–1.21). The subgroup analysis resulted significant for young women and middle aged men [126].

Chung’s study, neither included in the metanalysis, also assessed the relationship between urolithiasis and stroke. The design of the study was very similar to that of Kim´s study, but was a cohorts study with a follow up of five years. After adjusting for hypertension, diabetes, hyperlipidemia, obesity, gout cardiovascular diseases and urbanization level, urolithiasis remained as a significant risk factor for the development of stroke (HR: 1.43; 95% CI 1.35–1.50, *p* < 0.0001) [127].

Aydin et al. performed a single center, cross sectional, case control study conducted on 200 consecutive patients with CaOx stone disease, age and sex matched with 200 controls without any acute or chronic disease. Results showed that patients with CaOx stones had a higher Framingam Risk score (OR: 8.36; 95% CI 3.81–18.65, *p* < 0.0001) and SCORE risk score (OR: 3.02; 95% CI 1.30–7.02) compared with controls. The results also indicated that patients with urolithiasis had higher levels of total cholesterol (*p* < 0.0001), lower levels of HDL-cholesterol (*p* < 0.0001), and higher systolic blood pressure (*p* < 0.0001) [128].

In another case control study that examined the impact of CHD risk factors on CaOx stone formation, it was concluded that CaOx formers are significantly associated with several CHD risk factors, including smoking habit, hypertension, hypercholesterolemia, and obesity [129].

Arterial structure has also been evaluated in nephrolithiasic patients. Kim et al. demonstrated the association of coronary artery calcification with urolithiasis, assessed by cardiac computed tomography [130], while Fabris et al. confirmed an increased arterial stiffness in a study conducted on 42 idiopathic calcium stone formers [131]. Furthermore, the calcium sensing receptor, a G protein-coupled receptor, that is expressed in calcitropic tissues as the parathyroid gland and the kidney has been postulated to be the connection between nephrolithiasis and vascular calcification [132].

On the other hand, three studies have assessed endothelial function in kidney stone formers by using the Celermajer method, by measuring the flow-mediated dilation (%FMD), as an endothelium dependent response to shear stress [98]. All of them showed a significant lower %FMD in lithiasic patients [7,8,133]. No longitudinal cohort studies have been conducted on lithiasic patients with normal endothelial function in order to establish whether nephrolithiasis can be considered as a prior risk factor for ED.

A significant relationship between urolithiasic risk factors such as urine calcium/creatinine index and %FMD has also been reported [8]. On the other side, Saenz et al. studied the behavior of inflammation, OS and ED serum markers (CRP, IL6, MDA, VCAM-1 and ADMA) in relation to %FMD in lithiasic patients, but no relationship could be demonstrated [7].

Evaluation of stone and urine composition has been another approach used for the study of the relationship between cardiovascular disorders and kidney stone formation. Barbagli et al. have recently examined the composition of the stones and the urine metabolic profile of lithiasic patients with and without CVD (MI, angina, coronary revascularization or surgery of calcified heart valves). Patients with CVD had significantly lower 24 h urinary excretions for citrate and magnesium (both of them kidney stone risk factors). No differences in stone composition were found between groups [134].

Proteomic analysis of urine in pediatric stone formers and in relation to patients’ kidney stone risk factors, including hypercalciuria and hypocitraturia, has also been performed. Most of the proteins upregulated in both groups were involved in coagulation, fibrinolysis, metabolism and OS, all of them CVD risk factors. These results also suggest shared risk factors between urolithiasis and CVD through the proteomic urine analysis [135].

Hyperoxaluria has also been reported also to cause systemic ED in animal models and cell culture studies. Induction of hyperoxaluria in a rat model has been related to an increased systemic and renal tissue ADMA. ADMA is an endogenous competitive inhibitor of NOS, linked to ED and atherosclerosis. These results again suggest the interaction between the renal epithelial injury triggered by the hyperoxaluria, mediated by OS, and ED. The close contact between proximal tubular cells and endothelial cells makes hyperoxaluria affect the capacity of synthetizing vasoactive substances, growth modulators and inflammatory factors by endothelial cells [6] (Figure 1). Another cell culture study with co-incubation of endothelial and epithelial cells has suggested the protective role of endothelial cells against the epithelial tubular cell injury, induced by hyperoxaluria, by decreasing OS in renal tissue with potent antioxidants, which antagonize ADMA, and restores endothelial function [136]. 

**Table 1 ijms-23-00912-t001:** Characteristics of metanalysis and clinical studies in which association between urolithiasis and cardiovascular morbidity has been studied. CHD: Coronary heart disease. CAD: Coronary artery disease. MI: Myocardial infarction.

Type of Study	Number of Patients	Pathological Condition	Association	Reference
Metanalysis	3,658,360 participants 157,037 urolithiasis	CHD and stroke	CHD: RR = 1.24 (95% CI 1.14–1.36) MI: RR = 1.24 (95% CI 1.10–1.40) Stroke: RR = 1.21 (95% CI 1.06–1.38)	Peng and Zheng, 2017 [123]
Metanalysis	34,244,855 controls 52,791 urolithiasis	CAD	RR = 1.24 (95% CI 1.10–1.40) Women: RR = 1.43 (95% CI 1.12–1.82) Men: RR = 1.14 (95% CI 0.94–1.38)	Cheungpasitporn et al., 2014 [125]
Metanalysis	3,558,053 controls 49,597 urolithiasis	CAD Stroke MI Coronary revascularization	CAD: RR = 1.19 (95% CI 1.05–1.35) Stroke: RR = 1.40 (95% CI 1.20–1.64) MI: RR = 1.29 (95% CI 1.10–1.52) CR: RR = 1.31 (95% CI 1.05–1.65) Male cohorts no association	Liu et al., 2014 [124]
Cohorts Longitudinal	90,544 controls 22,636 urolithiasis	Ischemic Stroke	HR = 1.13 (95% CI 1.06–1.21)	Kim et al., 2019 [126]
Cohorts Longitudinal	5571 urolithiasis	Atherosclerotic cardiovascular disease	OR: 1.03 (95% CI 1.06–1.21) No association	Glover et al., 2016 [137]
Cohorts Longitudinal	40,773 controls 40,773 urolithiasis	MI Stroke Cardiovascular events	M.I: HR: 1.31 (95% CI 1.09–1.56) Stroke: HR: 1.39 (95% CI 1.24–1.55) CE: HR: 1.38 (95% CI 1.25–1.51)	Hsu et al., 2016 [138]
Cohorts Longitudinal	3,195,452 controls 25,532 urolithiasis	MI Revascularization	MI: HR = 1.40 (95% CI 1.30–1.51) Revasc.: HR = 1.63 (95% CI 1.51–1.76) Stroke: HR = 1.26 (95% CI 1.12–1.42)	Alexander et al., 2014 [139]
Self-reported history of urolithiasis Longitudinal	196,357 women 45,748 men 19,678 urolithiasis	MI Revascularization	HR = 1.30 (95% CI 1.04–1.62) (women) HR = 1.26 (95% CI 1.07–1.55) (women)	Ferraro et al., 2013 [140]
Cohorts Longitudinal	125,905 matched controls 25,181 urolithiasis	Stroke	HR = 1.43 (95% CI 1.35–1.50)	Chung et al., 2012 [127]
Cross sectional case-control study	200 CaOx stone formers	Cardiovascular disease Mortality	OR = 8.36 (95% CI 3.81–16.85) OR = 3.02 (95% CI 1.30–7.02)	Aydin et al., 2011 [128]
Self-reported history of urolithiasis Cross sectional	23,346	MI Stroke	OR = 1.34 (95% CI 1–1.79) OR = 1.33 (95% CI 1.02–1.74)	Domingos and Serra, 2011 [56]
Cross sectional study Retrospective cohort	10,860 control 4564 cases	MI	HR = 1.31 (95% CI 1.02–1.69)	Rule et al., 2010 [141]
Cross sectional study Retrospective cohort	Females cohort of 9704 426 nephrolithiasis	MI	RR = 1.78 (95% CI 1.22–2.62)	Eisner et al., 2009 [142]
Cohorts Longitudinal	10,938 normal blood pressure individuals 56 strokes	Stroke	3.6% urolithiasisin stroke patients 0.5% urolithiasis in no stroke individuals	Li et al., 2009 [143]
Retrospective cohort	1316 cases	MI		Westlund et al., 1973 [144]
Cross sectional study	2000 men	CHD	No association	Ljunghall et al., 1976 [145]
Cross sectional study	13,418 control men 404 current kidney stone formers 1231 past kidney stone formers	Risk factors for coronary disease: BMI, hypertension, diabetes mellitus, hyperuricemia, dyslipidemia and chronic kidney disease	Multivariate adjusted OR significant: Overweight/obesity (*p* < 0.001), hypertension (*p* < 0.001), gout/hyperuricemia (*p* < 0.001), chronic kidney disease (*p* < 0.001) in past and current kidney stone formers	Ando et al., 2013 [146]
Cross sectional case-control study	187 Controls 181 CaOx stone formers	Risk factors for coronary disease: BMI, hypertension, diabetes mellitus, hyperuricemia, hypercholesterolemia, smoking habit and current alcohol use	Multivariate adjusted OR significant: smoking habit (OR: 4.41, *p* < 0.0001), hypertension (OR: 3.57, *p* < 0.0001), hypercholesterolemia (OR: 2.74, *p* < 0.001), chronic kidney disease (*p* < 0.001)	Hamano et al., 2005 [129]
Cross sectional study	27 control 27 lithiasic patients	Endothelial dysfunction (Celermajer method)	%FMD 11.85% (SE: 2.78 lower in lithiasic patients. *p* < 0.001	Sáenz-Medina et al., 2021 [7]
Cross sectional study	1.-MetS- SD−: 93 2.-MetS- SD+: 93 3.-MetS+ SD−: 93 4.-MetS+ SD+: 93	Endothelial dysfunction (Celermajer method)	1 vs. 2: *p* < 0.001 1 vs. 3: *p* < 0.001 1 vs. 4: *p* < 0.001	Yazici et al., 2020 [133]
Cross sectional study	60 control 60 lithiasic patients	Endothelial dysfunction (Celermajer method)	FMD% Lithiasis: 6.49 ± 3.52 Control: 10.50 ± 5.10 *p* < 0.0001	Yencilek et al., 2017 [8]

## 8. Concluding Remarks

Nephrolithiasis is a worldwide public health disorder of increasing incidence, especially in industrialized countries in which diet factors have been pointed out as responsible of the higher urolithiasis prevalence. Cardiovascular disorders have been independently associated to urolithiasis disorders and lithiasic patients present a higher incidence of cardiovascular diseases, myocardial infarction and stroke with relative risks between 1.20 and 1.24.

Oxidative stress has been pointed out as an important factor in lithogenesis and a trigger factor for the vascular complications of low-chronic inflammation diseases such as obesity, diabetes or hypertension. Endothelial dysfunction is the preclinical stage of atherosclerosis and is also highly associated with cardiovascular morbidities. Oxidative stress and inflammation play an important role in the pathogenesis of endothelial dysfunction. In addition, endothelial dysfunction has been related to urolithiasis, so it may be considered as an intermediate and changeable feature between urolithiasis and cardiovascular disorders.

Special attention must be paid for some concomitant comorbidities associated with urolithiasis. Active treatment of hypertension or hyperlipidemia with angiotensin receptor blockers or statins allows to take advantage to prevent endothelial dysfunction through their pleiotropic effects and prevent the cardiovascular morbidity. On the other side, further studies are needed to clarify the mechanisms linking urolithiasis and endothelial dysfunction evidenced by epidemiological studies.

## Figures and Tables

**Figure 1 ijms-23-00912-f001:**
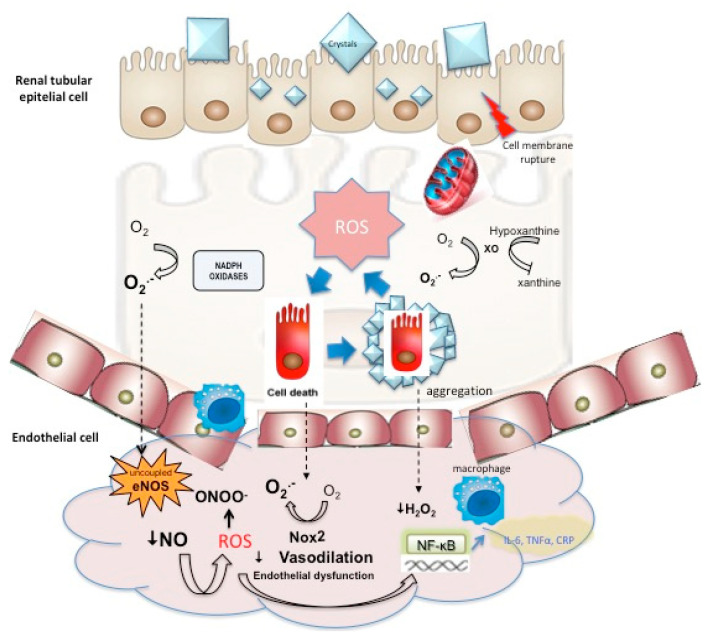
Interaction between oxidative stress-mediated epithelial cell injury, cell death, and stone formation with endothelial dysfunction triggered by impaired endothelium-dependent vasodilation augmented ROS and inflammatory response.

## Data Availability

Not applicable.

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
