# Peer review of "Endothelial Dysfunction: An Intermediate Clinical Feature between Urolithiasis and Cardiovascular Diseases"

_ijms, 2022, doi:10.3390/ijms23020912_

Round 1

Reviewer 1 Report

The review is very interesting, cardiovascular diseases constitute the major cause of death worldwide and Endothelial Dysfunction is one of the main causes. However, it needs a major overhaul before publication, especially in terms of sequencing.

General comments:

  1. The abstracted is good.
  2. The introduction could be improved, it should define and explore a little what is an endothelial dysfunction (please see the review Implications of Endothelial Cell-Mediated Dysfunctions in Vasomotor Tone Regulation, https://doi.org/10.3390/biologics1020015) and the Urolithiasis (Polycystic kidney disease, DOI: 10.1038/s41572-018-0047-y; Crosstalk between Renal and Vascular Calcium Signaling: The Link between Nephrolithiasis and Vascular Calcification, DOI: 10.3390/ijms22073590).
  3. The table 1 are the clinical studies or Metanalysis? I do not understand, why the author considered a Metanalysis as a clinical study?
  4. The authors say in the conclusion “Active treatment of hypertension, hyperuricemia or hyperlipidemia with angiotensin receptor blockers, allopurinol or statins allows to take advantage to prevent endothelial dysfunction through their pleiotropic effects and prevent the cardiovascular morbidity.”
  5. However, it only talks about these drugs in the summary and in the discussion, it does not say their function, it does not explore these pathologies from a therapeutic point of view. I think it would be interesting to explore this part, even making a chapter on the subject.
  6. The Figure 1 is neither explained nor explored in the article, and is a very important point of the article.
  7. The section 3. Ros and Renal Oxidative Stress Sources, needs to be introduced.
  8. The sequence of the objects in my opinion is not the best, because it would be more logical to first mention items 5 and 6 and then 3 and 4.

Author Response

Thank you very much for your valuable suggestions, all of them have been taken in to account and discussed sin the attached file in a detailed manner.

Reviewer 2 Report

The authors has written a review on the association between urolithiasis and cardiovascular diseases considering endothelial dysfunction.

The topic of review is interesting, and they summarized a lot of evidence on this topic. However, there are several concerns in this manuscript.

Firstly, the location of Table 1 should be Chapter 7 because they explained the previous studies there.

Secondly, there are several typos in using Abbreviations. For example, Nox, NFkB, Ros, and PgI2 should be corrected. In addition, “et al” should be “et al.” .

Thirdly, the numbers of references should be merged properly.

Fourth, it would be better for the authors to use “cardiovascular diseases” throughout the manuscript because they included studies focusing on various cardiovascular diseases. In addition, they used the cardiovascular “disorders” at the beginning of this manuscript.

Author Response

Thank you very much for your valuable comments, all of your suggestions have been included in the manuscript.

Round 2

Reviewer 1 Report

The article has been improved, however in my opinion the articles referred to in the previous point 2 would be an asset to this review, since this review presents a reduced amount of recent articles, out of 147 articles only 22 are from the last 5 years.

  • Endothelial Cell-Mediated Dysfunctions in Vasomotor Tone Regulation, https://doi.org/10.3390/biologics1020015;
  • Polycystic kidney disease, DOI: 10.1038/s41572-018-0047-y;
  • Crosstalk between Renal and Vascular Calcium Signaling: The Link between Nephrolithiasis and Vascular Calcification, DOI: 10.3390/ijms22073590.

Also, there are some poorly formatted references, such as

9- Yazici O, Narter F, Erbin A, Aydin K, Kafkasli A, Sarica K. Effect of endothelial dysfunction on the pathogenesis of urolithiasis in patients with metabolic syndrome. Aging Male. 20191-6.

  1. Cachofeiro V, Goicochea M, de Vinuesa SG, Oubiña P, Lahera V, Luño J. Oxidative stress and inflammation, a link between chronic kidney disease and cardiovascular disease. Kidney Int Suppl. 2008S4-9.
  2. Bargagli M, Moochhala S, Robertson WG, Gambaro G, Lombardi G, Unwin RJ et al. Urinary metabolic profile and stone composition in kidney stone formers with and without heart disease. J Nephrol. 2021

Author Response

Thank you for your valuables comments. All cites have been included and illustrated in the manuscript, and all cytes have been revised and correctly formatted.

Regards

Javier Saenz
